Gut bacterial communities in the freshwater snail Planorbella trivolvis and their modification by a non-herbivorous diet

Hu Zongfu 1 2
http://orcid.org/0000-0002-6733-7340 Tong Qing 2
Chang Jie 1
Yu Jianhua 1
Li Shuguo 1
Niu Huaxin 1 niuhx@imun.edu.cn
Ma Deying 2 2967233453@qq.com
1 College of Animal Science and Technology, Inner Mongolia University for Nationalities , Tongliao , People’s Republic of China
2 College of Animal Science and Technology, Northeast Agricultural University , Harbin , China
Gillespie Joseph
Electronic publication date: 2021 Feb 12
Publication date: 2021
Volume: 9
Electronic Location ID: e10716
Received 2020 Jul 9; Accepted 2020 Dec 15
Copyright: © 2021 Hu et al.
Copyright year: 2021
Copyright holder: Hu et al.
License: This is an open access article distributed under the terms of the Creative Commons Attribution License, which permits unrestricted use, distribution, reproduction and adaptation in any medium and for any purpose provided that it is properly attributed. For attribution, the original author(s), title, publication source (PeerJ) and either DOI or URL of the article must be cited.
License URL: https://creativecommons.org/licenses/by/4.0/

Keywords: Planorbella trivolvis, Freshwater snails, Cellulose, Gut microbiota, Functional prediction, Intestine histology

Funding: National Natural Science Foundation of China 31860730 and 31460692 Mongolia Key Laboratory of Toxicology and Toxicology MDK2018031 and MDK2019077 Grassland Talents Project Q2017022 Inner Mongolia Natural Science Foundation Project MDK2019077 Scientific Research Fund Project of Inner Mongolia University for Nationalities NMDYB1705 This work was supported by the National Natural Science Foundation of China (31860730; 31460692), the Mongolia Key Laboratory of Toxicology and Toxicology (MDK2018031; MDK2019077), The “Grassland Talents Project” of Inner Mongolia Autonomous Region, the First Level Training Program for Young Innovative and Entrepreneurial Talents (Q2017022), the Inner Mongolia Natural Science Foundation Project (MDK2019077), and the Scientific Research Fund Project of Inner Mongolia University for Nationalities (NMDYB1705). The funders had no role in study design, data collection and analysis, decision to publish, or preparation of the manuscript.

==============================
The freshwater pulmonate snail Planorbella trivolvis is a common species in various bodies of water but is not native to China. Planorbella trivolvis usually live on diets with high fiber content, such as water grasses, algae and fallen leaves. These snails can attach to the wall of a water tank or to water grass and can be transported overseas to China through the ornamental fish trade. There are few studies investigating the intestinal microbiota of freshwater snails. In this study, using culture-independent molecular analysis, we assessed for the first time the complexity of bacterial communities in the intestines of reared snails. The intestinal microbiota in the snails fed different diets, that is, herbivorous feed (HV) with high cellulose and non-herbivorous feed (NHV) with low cellulose, were analyzed by Illumina sequencing. The results showed that the NHV-based diet significantly increased the body mass, shell diameter and specific growth rate of the snails after 60 days of rearing (P < 0.05). Histological experiments showed that the fat droplets in the epithelium columnar cells of the intestines of the NHV snails increased, and the cilia on these cells fell off. The sequencing results identified 486 and 195 OTUs in HV and NHV, respectively. Lots of bacteria were not reported previously in snails. The intestinal microbiota diversity index (Shannon, Simpson, Ace and Chao) in the NHV snails was significantly lower than that in the HV snails. The gut microbiota in the HV snails were predominantly Proteobacteria (52.97%) and Bacteroidetes (28.75%), while the gut microbiota in NHV snails were predominantly Proteobacteria (95.23%). At the genus level, Cloacibacterium (24.60%), Pseudomonas (4.47%), OM6ON (6.12%), and Rhodobacter (5.79%) were observed to be abundant in HV snails. However, Aeromonas (85.4%) was determined to be predominant in NHV snails. Functional prediction of the gut microbiome in snails by PICRUSt demonstrated a significant difference between the two groups, and the HV snails exhibited higher lignocellulose enzyme activity than did the NHV snails. This study represents a first step in characterizing the gut microbiota of the freshwater snail. Most of these microbes can process plant biomass and digest cellulose and lignocellulose, and the enzymes of these bacteria may have potential biotechnological applications in a variety of industrial processes.

Introduction

Planorbella trivolvis belongs to the family Plamorbidae and is primarily distributed in Africa, Europe and North America (Hunter, 1990). Planorbella trivolvis is one of the most abundant taxa in the subclass Pulmonata, with more than 250 species in 40 genera being recorded worldwide. These species can serve as intermediate hosts for a variety of trematodes, such as Ribeiroia ondatrae and echinostome, thereby contributing to the global disease burden (Klockars, Huffman & Fried, 2007; Peterson, 2007). Snails are common in ponds, lakes and marshes and feed on algae, aquatic plants, the fallen leaves of terrestrial plants and various types of detritus (Lombardo & Cooke, 2002). Therefore, the ability of snails to digest lignocellulose is notable, and they are thought to contain a gut microbiome that is specialized in the rapid hydrolysis and fermentation of lignocellulosic plant biomass (Pinheiro et al., 2015; Wijanarka, Kusdiyantini & Parman, 2016).

There is considerable bacterial diversity in the intestinal tracts of animals. These bacteria can interact with host animals, enhance animal immunity, help to digest nutrients, and play important roles in inhibiting harmful foreign bacteria (Lee et al., 2013; Carnevali, Maradonna & Gioacchini, 2017; Guarner & Malagelada, 2003). Up to now, the gut microbiota in herbivores have been well studied in many animals, such as cows, ants, and pandas (Bergmann et al., 2015; Jami et al., 2013; Russell et al., 2009; Tun et al., 2014).

However, the gut microbial communities of snails remain largely unstudied (Joynson et al., 2017; Van Horn et al., 2012). The early study of intestinal microbiota in snails employed culture-based methods, and a limited number of bacteria were identified, such as Enterobacter, Enterococcus, Lactococcus, and Clostridium (Charrier et al., 2006). Next, molecular biology techniques were used to study the snail gut microbiota, such as DGGE fingerprint analysis, which has been utilized to obtain the structural composition of the dominant bacterial community in the snail Helix pomatia, and the number of identified bacteria was also limited (Nicolai et al., 2015). Recently, high-throughput sequencing based on parts of 16S rRNA gene, which can reveal more kinds of microbiota even that with low abundance, were used to study the gut microbiomes in freshwater snails Biomphalaria pfeifferi, Bulinus africanus, Helisoma duryi and Radix ovata (Van Horn et al., 2012; Hu et al., 2018). Furthermore, the intestinal microorganisms of snail play an important role in cellulose decomposition. Some studies have isolated cellulolytic bacteria from the intestine of snails by carboxymethylcellulose (CMC) plates (Charrier et al., 2006; Pinheiro et al., 2015). Other studies, using metagenomic, have identified a great number of genes involved in lignocellulosic breakdown in intestinal microorganisms of snails Achatina fulica, Arion ater (Cardoso et al., 2012; Joynson et al., 2017).

The main focus of this work was to characterize intestinal microbiomes of Planorbella trivolvis shaped by diets. Previous studies showed that high-fibre or low-fibre diets shape the intestinal microbiota and microbial metabolites and affect the growth performance and intestinal morphology in various animals, such as pigs and rats (Stark, Nyska & Madar, 1996; Heinritz et al., 2016; Coble et al., 2018). Therefore, the variation in fiber content in diets may affect the intestinal microbiota in snails or affect the growth and intestinal status of snails. However, the effects of specific dietary modulation on the intestinal microbiota and growth of snail have not been fully elucidated.

In this study, a culture-independent molecular analysis based on the 16S rRNA gene was performed (1) to demonstrate the effect of the diet (high-cellulose and low-cellulose) on gut microbiota in Planorbella trivolvis and (2) to examine functional differences in the microbiomes of snails fed different diets. The effects of diet on snail growth and intestinal morphology were also examined in this study.

Materials and Methods

Rearing of snails

The snails were originally purchased from the ornamental fish market, and a breeding colony were formed in the laboratory. When a sufficient number of newborn snails were reared to about 5–7 mm of shell diameter, a total of 120 juvenile snails with uniform size (5.88 ± 0.14 mm of shell diameter) were selected and randomly distributed into two groups by their diets: herbivorous groups (HV) (60 snails) and non-herbivorous groups (NHV) (60 snails). Each group had three replicate aquariums with 20 snails in each repetition. HV group was fed dry alfalfa contain a high content of cellulose (26%), and NHV group was fed market-purchased pellet feed (Takara sakana-ii, for ornamental fish feeding), which contained low cellulose (5%) (Table 1). The aquaculture water temperature was maintained at 24~25 °C, with a pH of 6.7 ± 0.4. The snails were reared for 60 days and no snails died during rearing.

Table 1 Chemical characteristics of alfalfa (g/100 g).

Diets type	Protein	Crude fiber	Crude fat	Crude ash	
HV (alfalfa)	19.62 ± 0.23	26.75 ± 0.24	4.86 ± 0.11	8.34 ± 0.02	
NHV (pellet feed)	27.22 ± 0.09	4.97 ± 0.03	2.17 ± 0.05	9.03 ± 0.06	
Note:

HV (alfalfa): the herbivorous snails reared with alflafa; NHV(pellet feed): the non-herbivorous snails reared with pellet feed. The data on the composition of the pellet feed come from the manufacturer (Takara sakana-ii). The composition of the alfalfa was determined in laboratory as follwing methods: (1) The NDF and ADF levels were measured as described by Van Soest, Robertson & Lewis (1991). (2) The crude protein (CP) content was measured by the Kjeldahl N method (AOAC 2000). (3) Soxhlet immersion extraction was used for measure crude fat. (4) Crude ash was measured by firing the sample for 30 min under 550 °C in muffle furnace.

Sample collection and processing

After 60 days of rearing, the body length and weight of snails were determined first. Next, the intestines were sampled. When the intestines were collected, the snails were cleaned with tap water, and the surface of the shell was subsequently wiped with 70% alcohol. The shells were carefully broken and disassembled from the snail body. After that step, the intestines were extracted carefully to avoid rupturing the gut wall. Next, the intestines were sampled and contained their contents. All dissections were conducted under aseptic conditions. For histological examination, the collected intestines were fixed in 4% formaldehyde for histological examination. Because the intestines of individual snails were too small to meet the sampling requirements, the intestines and contents of the 5–8 snails were packed into a centrifuge tube to form a repeat sample for sequencing.

Growth index

The growth performance of snails after 60 days of rearing was characterized by their weight gain rate (WGR) and specific growth rate (SGR). The WGR was calculated by comparing their weight gain (WG) with their initial average weight (IW). The SGRs were obtained by dividing their weight gain by 60 days. The formulas of WGR and SGR are as follows:

WGR (%) = [(TW − IW)/IW] × 100

SGR (%/d) = [(TW − IW)/d] × 100

WGR: Weight gain rate; SGR: Specific growth rate; TW: terminal weight; IW: initial weight; d: rearing days.

Intestinal morphology analysis

To examine the effect of diets on the intestinal morphology of snails, the intestinal samples were washed in saline solution and fixed in 4% paraformaldehyde solution above 48 h. Next, the fixed samples were successively treated as follows: dehydrated by an alcohol gradient (50%, 70%, 90% and 100%), xylene transparency, and paraffin embedding. Three paraffin sections from each sample were cut and stained by the hematoxylin-eosin (HE) method. Last, the stained slice was sealed by neutral gum and subjected to observation and scenery shooting using a panoramic scanner (3DHISTECH; Pannoramic MIDI, Budapest, Hungary).

Determination of intestinal microbiota diversity

The intestine samples obtained as described above contained the contents used for genome extraction. Total DNA was extracted using the FAST DNA™ Spin Kit for soil (MP-BIO, Santa Ana, CA, USA) following the protocol of the manufacturer. The extracted DNA was subjected to 1.5% agarose-gel electrophoresis, and the DNA concentration and OD260 nm/OD280 nm (OD, optical density) value were determined with a Nanodrop spectrophotometer (Thermo Scientific, Wilmington, DE, USA).

To detect intestinal bacteria, the V3~V4 regions of the bacterial 16S rRNA gene were amplified with DNA primers 338F (5′-ACTCCTACGGGAGGCAGCAG-3′) and 806R (5′-GGACTACHVGGGTWTCTAAT-3′) (Mori et al., 2013). The 12 bp barcodes on both ends of the primers were used to identify the sequences of different samples. Next, PCR amplification was conducted, and the procedure was as follows: initial denaturation at 95 °C for 3 min followed by 27 cycles of denaturing at 95 °C for 30 s, annealing at 55 °C for 30 s and extension at 72 °C for 45 s, single extension at 72 °C for 10 min, and ending at 4 °C. The amplified products were subjected to electrophoresis and extracted from a 2% agarose gel. Next, the products were purified using the AxyPrep DNA Gel Extraction Kit (Axygen Biosciences, Union City, CA, USA). After Qubit quantification and detection, amplicons from each PCR sample were normalized to equimolar amounts, and high-throughput sequencing was performed using the 2 × 300 bp protocol on an Illumina MiSeq PE300 platform (Illumina, San Diego, CA, USA).

The raw reads were deposited into the NCBI Sequence Read Archive (SRA) database (Accession Number: PRJNA640745).

Raw fastq files were demultiplexed, quality-filtered by Trimmomatic (Bolger, Lohse & Usadel, 2014) and merged by FLASH. UPARSE (Edgar, 2013) was utilized to conduct operational taxonomic unit (OTU) clustering analysis at 97% identity, and chimaeric sequences were identified and removed using UCHIME (Edgar et al., 2011). Next, the representative OTUs were analyzed on the Qiime platform (Caporaso et al., 2010) against the Silva_132 16S Database (http://www.arb-silva.de) to determine taxonomy.

The heatmap analysis was performed in the heatmap package in R. Venn and PCoA analyses were performed in the vegan package in R. The alpha-diversity of the gut bacterial community, containing indices of Sobs, Shannon, Chao, Simpson, Ace, and coverage, was analyzed by Mothur (Schloss et al., 2009). Biomarker discovery was performed on the linear discriminant analysis (LDA) effect size (LEfSe) to identify the specific organisms whose relative abundances differ between two groups of samples (Segata et al., 2011).

To examine functional differences in the microbiome of snails, the functional attributes of metabolic genes from the snail gut microbiota (KEGG Orthologs, KOs) were predicted by a phylogenetic investigation of communities by reconstruction of unobserved states (PICRUSt), which were obtained by macthing the sequencing data against the genomic KEGG database (Langille et al., 2013). As a computational approach, PICRUSt can predict the functional composition of an obtained metagenome using marker gene data and a database of reference genomes (Langille et al., 2013).

Statistical analysis

SPSS 19.0 software was employed for a two-tailed Student’s t-test on the growth data of snails. P values < 0.05 were considered to indicate significant differences, and descriptive statistics were expressed as the mean ± SD.

Results

Snail growth

The initial size and weight did not differ between the two groups (P > 0.05) (Table 2). After 60 days of rearing, the diameter of the final shell of the HV snails was about 10.53 mm, representing an increase of 176.08% compared with the initial shell diameter (Table 2). The final shell diameter of NHV snails was about 11.86 mm, representing an increase of 205.19% compared with the initial shell diameter; thus, and the final shell diameter of snails in the NHV group was significantly larger than that of the HV group (P < 0.05). The endpoint weight of HV group was 0.219 g, and that of NHV group was 0.432 g. The weight gain rate of the NHV group (566.90%) was significantly higher than that of the HV group (321.40%) (P < 0.05). Also, the specific growth rate of snails in the NHV group (3.16%) was significantly higher than that in the HV group (1.94%) (P < 0.05) (Table 2).

Table 2 Growth performance of Planorbella trivolvis.

Items	Groups	Initial value	Difference	Final value	Difference	
Diameter of shell/mm	HV	5.98 ± 0.07	a	10.53 ± 0.45	b	
	NHV	5.78 ± 0.13	a	11.86 ± 0.11	a	
Weight/g	HV	0.068 ± 0.003	a	0.219 ± 0.003	b	
	NHV	0.065 ± 0.005	a	0.432 ± 0.007	a	
WGR/%	HV	\		321.40 ± 16.01	b	
	NHV	\		566.90 ± 46.28	a	
SGR/%	HV	\		1.94 ± 0.08	b	
	NHV	\		3.16 ± 0.11	a	
Note:

In the same column, values with no letter or the same letter superscripts mean no significant difference (P > 0.05), while with different letter superscripts mean significant difference (P < 0.05). HV, the herbivorous snails reared with alflafa; NHV, the non-herbivorous snails reared with pellet feed. SGR, special growth rate; WGR, Weight gain rate.

Intestinal morphology

To examine the effect of feed on the intestinal morphology of snails, histological examination were conducted (Fig. 1). The differences between treatment groups were clearly shown in the simple columnar epithelium cells (SCEC). For the HV snails, the intestinal structure was intact, and the SCEC arrangement was regulated. The microvilli were tightly attached to the SCEC. For the NHV snail, the intestinal SCECs were filled with fat droplets and thus appear transparent in this image (Figs. 1C and 1D). The fat droplets occupied the cytoplasm of the SCEC, and the nucleus was observed to be indented (Figs. 1C and 1D), indicating that low fiber feeding leads to increased fat deposition in intestinal epithelial cells. The microvilli were detached to a certain extent from the SCEC.

Figure 1 Effects of different diets on intestinal histology of Planorbella trivolvis.

(A) Intestinal morphology of HV snails (200 µm); (B) Intestinal morphology of HV snails (50 µm); (C) Intestinal morphology of NHV snails (200 µm); (D) Intestinal morphology of NHV snails (50 µm).

Alpha diversity based on 16S rRNA gene sequencing

A total of 737,284 raw reads were generated using the Illumina MiSeq sequence platform and 485,342 high quality sequences were obtained (following quality control and sequence filtration) with an average length of 459.9 bp. The mean number of sequences per sample was 54,277.60 ± 6,298.01 in HV snails and 39,606.60 ± 6,846.06 in NHV snails (Table 3). The 31,486 rarefied sequences were clustered into 525 OTUs at 97% identity, with 486 and 195 OTUs being identified in HV and NHV, respectively.

Table 3 Alpha-diversity indices of intestinal microbial flora of Planorbella trivolvis.

Estimators	HV-Mean	HV-Sd	NHV-Mean	NHV-Sd	P-value	
OUT number	307.60	25.71	90.00	46.67	0	
Sequences number	54277.60	6298.01	39606.6	6846.06	0.008	
sobs	305.2	24.56	87.00	46.67	1.513e−05	
shannon	3.42	0.19	1.00	0.50	7.253e−06	
simpson	0.09	0.03	0.61	0.17	0.000133	
ace	379.08	29.34	118.69	47.56	6.243e−06	
chao	375.04	26.28	105.4	47.40	3.809e−06	
coverage	0.9976	0.0003	0.9992	0.0003	4.689e−05	
Note:

HV, the herbivorous snails reared with alflafa; NHV, the non-herbivorous snails reared with pellet feed. P-values were from a t-test.

The obtained 525 OTUs include members of 419 species, 304 genera, 184 families, 108 orders, 37 classes and 19 phyla. There were 18 phyla, 286 genera, 108 orders, 36 classes, 486 OTUs belonging to HV, and there were 13 phyla, 52 orders, 21 classes, 132 genera, and 195 OTUs belonging to NHV. All of the alpha-diversity indices (Sobs, Shannon, Simpson, Ace, Chao) were significantly different between HV and NHV populations (P < 0.05) (student’s t-test) (Table 3). The Good’s coverage for the observed OTUs was above 99.70%, and the rarefaction curves showed clear asymptotes (Fig. 2A), which, taken together indicated that the given level of sampling effort was sufficient to capture the bacterial communities in snails. Notably, there were clear differences for the rarefaction and rank-abundance curves between the treatment groups (Figs. 2A and 2B). The abundance analysis showed that there were only 5 OTUs in the NHV snails with abundances greater than 1%, while there were as many as 22 OTUs in the HV snails (Fig. 2B). The total abundance of OTUs with abundance greater than 1% reached 79.68% in the HV snails and 83.39% in the NHV snails. The number of core OTUs identified in all samples was 22 (Fig. 2C).

Figure 2 Diversity of gut community in snails fed with different diets types.

(A–C) Rarefaction curves (A), Rank-abundance curves (B), and core analysis (C) of intestinal microbial flora at OTU level; (D–F) VENN analysis of gut community at phylum (D), family (E), and genus (F) level.

Venn analysis showed that the bacteria belonging to 12 phyla, 77 families, and 114 genera were shared by two groups: 6 phyla, 97 families, and 172 genera were unique in HV snails, and 1 phylum, 10 families, and 18 genera were unique in NHV snails (Figs. 2D and 2F).

Composition of snail intestinal microbiota

The phyla Proteobacteria (52.97%) and Bacteroidetes (28.75%) were predominant among the bacteria in the HV group (Fig. 3A). The phyla with abundances above 1% were Actinobacteria, Verrucomicrobia, Chloroflexi, Cyanobacteria, and Chlamydiae. However, only Proteobacteria (95.23%) was predominant among the bacteria in NHV snails. Bacteroidetes was the other phylum with an abundance above 1%.

Figure 3 Composition of intestinal microbial flora at phylum (A), family (B) and genus (C) level.

The bacterial taxon with abundance below 1% in phylum, 4% in family, and 3% in genus level was merged and shows as “others”.

At the family level, there were 20 families and 8 families with abundances above 1% in the HV and NHV snails, respectively. Weeksellaceae (24.6%) and Rhodobacteraceae (12.38%) showed high abundance in HV snails. Aeromonadaceae (85.40%) dominated the microbiota in NHV snails (Fig. 3B).

At the genus level, the composition of intestinal microbiota differed between the HV group and the NHV group (Fig. 3C). In the HV group, there were 22 genera with abundances above 1%. Cloacibacterium (24.60%) showed the highest abundance in the HV groups. Others, such as Pseudomonas (4.47%), OM6ON (6.12%), and Rhodobacter (3.78%) were also abundant. In the NHV snail, there were 7 bacteria with abundances higher than 1%. Aeromonas, exhibited the highest abundance, accounting for 85.40%.

The heat map analysis constructed from the top 35 abundant genera, reflected that the intestinal microbiota in the two groups was different (Fig. 4). The Wilcoxon rank-sum test showed that the abundance of 26 genera was significantly higher in HV snails than in NHV snails (P < 0.05). Only two genera (Aeromonas and Comamonas) with significantly higher abundances were observed in NHV snails.

Figure 4 Heat map showing the relative abundance of the 35 bacterial genus in snail.

The HV snails are shown by H1, H2, H3, H4, H5 and NHV snails are shown by N1, N2, N3, A4, N5. The red asterisk represented the significantly different genera among two groups, based on Wilcoxon rank-sum test, *P < 0.05, **P < 0.01.

Clustering of the snail gut bacterial community

Clustering analysis of intestinal microbiota associated with diet were performed. Principal coordinates analysis (PCoA) of sequencing data using pairwise weighted and unweighted UniFrac distances showed that the bacterial community structure of HV was different from that of NHV (Fig. 5). UniFrac clearly separated different microbiota efficiently by diet, demonstrating the importance of food as a driver of community composition in this freshwater snail.

Figure 5 Principal Coordinates Analysis (PCoA) plots in intestinal microbiota of snails at OTU level using pairwise unweighted (A) and weighted UniFrac distances (B).

The HV snails are shown by blue dot and NHV snails are shown by orange triangle. Adonis: P-value = 0.004, R-value = 0.4916 (A); P-value = 0.004, R-value = 0.9170 (B).

LEfSe analysis (the LDA threshold was 4) was used to screen out microorganisms with significant differences in the snail intestines of the two groups fed different diets. There were 11 bacterial genera that were significantly enriched in the intestinal samples of the HV group, and there was one genus (Aeomonas) that was enriched in the NHV group (Fig. S1). When the LDA threshold was set to 2, 114 bacterial genera were significantly distributed in the intestinal samples of the HV group, while only 4 bacterial genera were significantly distributed in the NHV group (Table S1).

Although the intestinal microbiota of snails was influenced by food type, we found that 27 genera existed as core microbiota of all samples in both groups. Among these genera, 19 were affiliated with Proteobacteria, 4 were affiliated with Bacteroidetes, 3 were affiliated with Actinobacteria, 2 were affiliated with Verrucomicrobia and 1 was affiliated with Chloroflexi (Table 4).

Table 4 Shared core bacteria among two groups.

Phylum	Family	Genus	HV	SD	NHV	SD	
Proteobacteria	Rhizobiaceae	Allorhizobium	0.07	0.03	0.91	1.38	
Bacteroidetes	Flavobacteriaceae	Flavobacterium	1.17	0.45	0.02	0.01	
Proteobacteria	Rhizobiaceae	norank	0.01	0.01	0.12	0.17	
Proteobacteria	Shewanellaceae	Shewanella	0.11	0.16	1.11	1.36	
Proteobacteria	Rhodobacteraceae	Rhodobacter	3.78	0.37	0.02	0.02	
Bacteroidetes	Weeksellaceae	Cloacibacterium	24.6	5.66	2.75	3.33	
Proteobacteria	Aeromonadaceae	Aeromonas	1.78	0.8	85.4	14.57	
Verrucomicrobia	Rubritaleaceae	Luteolibacter	1.53	1.25	0.3	0.27	
Proteobacteria	Reyranellaceae	Reyranella	1.21	0.34	0.13	0.15	
Bacteroidetes	Cytophagaceae	Cytophaga	0.07	0.07	0.07	0.08	
Proteobacteria	Sphingomonadaceae	Altererythrobacter	0.02	0.01	0.02	0.02	
Proteobacteria	Rhizobiales_Incertae_Sedis	unclassified	0.15	0.03	0.06	0.07	
Verrucomicrobia	Rubritaleaceae	Haloferula	0.05	0.05	0.6	0.59	
Bacteroidetes	Chitinophagaceae	Sediminibacterium	0.11	0.05	0.26	0.37	
Actinobacteria	Demequinaceae	unclassified	0.35	0.13	0.41	0.29	
Actinobacteria	Mycobacteriaceae	Mycobacterium	2.58	0.88	0.02	0.02	
Proteobacteria	Beijerinckiaceae	unclassified	0.07	0.07	0.68	0.98	
Proteobacteria	Xanthomonadaceae	Stenotrophomonas	0.06	0.02	0.05	0.08	
Proteobacteria	Enterobacteriaceae	Kluyvera	0.56	0.58	0.02	0.04	
Proteobacteria	Pseudomonadaceae	Pseudomonas	4.47	0.5	1.84	1.81	
Proteobacteria	Enterobacteriaceae	unclassified	1.85	0.82	0.02	0.03	
Actinobacteria	norank_o__PeM15	norank	4.75	2.27	0.06	0.05	
Chloroflexi	JG30-KF-CM45	norank	2.51	0.45	0.01	0.01	
Proteobacteria	Rhodobacteraceae	unclassified	5.79	0.91	0.25	0.28	
Proteobacteria	Rhizobiales_Incertae_Sedis	norank	6.81	1.27	0.15	0.2	
Proteobacteria	Beijerinckiaceae	Bosea	0.12	0.06	1.58	2.41	
Proteobacteria	Hyphomicrobiaceae	Hyphomicrobium	0.51	0.12	1.42	2.12	
Note:

HV, the herbivorous snails reared with alflafa; NHV, the non-herbivorous snails reared with pellet feed.

Predictive metagenomic profiling

Phylogenetic investigation of communities by reconstruction of unobserved states analysis was performed to generate a predictive functional profile to gain insight into the metabolic capacity of the enteric microbiome (Fig. 6). Based on this analysis, the relative abundance of several COG functions was significantly different between feeding regimens, such as the enrichment of genes involved in energy production and conversion, lipid transport and metabolism in HV, and the enrichment of genes involved in amino acid transport and metabolism, signal transduction mechanisms in NHV. No significant differences were observed for the relative abundance of genes involved in carbohydrate transport and metabolism, inorganic ion transport and metabolism, and transcription.

Figure 6 Predictive functional profiles generated from 16S rRNA gene sequences using PICRUSt.

Significant differences between HV and NHV snails were observed for several COG function (two-tailed Student’s t-test ; *P < 0.05, **P < 0.01). The HV group is represented by green bars (H) and the NHV group by red bars (N).

Because plant fiber degradation requires a diverse suite of enzymes for complete hydrolysis, the abundance of genes for several lignocellulose-active enzymes was also examined (Fig. 7). Eighteen enzymes determined to be more enriched in HV snails than in NHV snails, acting upon cellulose, hemicellulose, lignin, and cello-oligosaccharides, respectively. Six enzymes were more enriched in NHV than HV snails, acting upon hemicellulose, and cello-oligosaccharides, respectively (Fig. 7). Enzymes related to protein degradation were also analyzed. The result showed that a large number of genes related to aminopeptidase, dipeptidase, carboxypeptidase and endopeptidase were more enriched in HV snails than in NHV snails (Fig. 8).

Figure 7 Predictive KEGG Orthology (KO) of lignocellulose-active enzymes across diet types.

H1 to H5 (red color) indicates the sequence number of predicted genes among the HV snails, and N1 to N5 indicates the sequence number of predicted genes among the NHV snails. Each column represents a different entry in the greengene database, each with a unique K-number.

Figure 8 Predictive KEGG Orthology (KO) of enzymes related to protein degradation in snails.

H1 to H5 (red color) indicates the sequence number of predicted genes among the HV snails, and N1 to N5 indicates the sequence number of predicted genes among the NHV snails. Each column represents a different entry in the greengene database, each with a unique K-number. The differences of each KO genes were showed by an asterisk at the top of the plot. *P < 0.05, **P < 0.01.

Discussion

For invertebrates, the study of gut microbes remains an open question and warrants further research to characterize roles played by bacteria. In this work, we investigated the effect of diets, containing a high (herbivorous diets, HV) or low (non-herbivorous diets, NHV) cellulose, on gut bacterial community structure in Planorbella trivolvis. We also compared the effect of these diets on the growth and gut histological morphology. NHV significantly promoted the body length and body weight growth of snails, indicating that snails obtained more crude protein and energy and less crude fiber when feeding on pellet feed. However, for herbivorous feed, namely, alfalfa, the snails needs to use their radulas to scrape the plant ingredients; therefore, the feeding efficiency is considerably lower.

The two diets were observed to influence the intestinal morphology of snails. As the results showed, there were no fat droplets in the simple columnar epithelium cells (SCECs) of HV snails. While in NHV snails, the intestinal SCECs are filled with fat droplets. The reason for this phenomenon may lie in diets in NHV with low dietary fiber, which contain more soluble carbohydrates and would be easily digested and absorbed by snails. Furthermore, dietary soluble carbohydrates can promote intestinal lipid accumulation in aquatic organisms (Zhao et al., 2020; Castro et al., 2019). Therefore, the fat droplets occupied the cytoplasm of the SCECs, and the nucleus was indented, which may be related to a disturbed state of the normal metabolism of the cell and caused the detachment of microvilli from SCEC in NHV snails.

Bacteria in the digestive tract may have the ability to ferment cellulose and chitin to produce substances that snails can easily absorb, such as small molecules of organic acids and sugars (Speiser, 2001; Charrier et al., 2006). In this study, the bacterial community in HV snails was dominated by Proteobacteria, Bacteroidetes and Actinobacteria, indicating which bacteria may be related to plant digestion. In fact, previous studies on herbivores have observed that Proteobacteria are the dominant species in snails (Cardoso et al., 2012; Joynson et al., 2017; Nicolai et al., 2015). In this study, we found that Bacteroidetes and Actinobacteria were also associated with plant-eating gastropods. At the phylum level, the research results of Van Horn et al. (2012) on the three Planorbidae have some similarities with ours. Their study showed that Proteobacteria, Bacteroidetes, and Acidobacteria were present and dominant in the intestines of these snails. A large number of bacterial species belonging to Proteobacteria, Bacteroidetes, and Acidobacteria are also observed in the gut of termites, which eat lignocellulose entirely (Mikaelyan, Meuser & Brune, 2017). However, for the American bison, a herbivore, the gut microbiota was very different from that of the freshwater gastropods, with the order of abundance from most to least being Firmicutes (53%), Bacteroidetes (33%) and Tenericutes (4%) (Bergmann et al., 2015).

Core microbes are microbes that are consistently present in a particular habitat, such as intestines of animals, despite the high variable conditions in that habitat. In this study, 27 genera were observed as core microbiota between the two dietary groups, among which Cloacibacteria, Aeromonas and Pseudomonas had the abundance above 1% in both groups (HV and NHV), and Aeromonas, Pseudomonas, Flavobacterium, and Enterobacteriaceae families have been characterized as common in terrestrial or aquatic gastropods by previous studies (Cardoso et al., 2012; Joynson et al., 2017). Although the role of the core microbiota in the gut microbiota of animals is not well understood, it is likely critical and needs further study (Shade & Handelsman, 2012; Kokou et al., 2019).

Some bacteria can be found both in the gut microbiota of snails and free-living in the natural water environment. Cloacibacterium, which is abundant in HV snails, is a facultative anaerobic and gram-positive bacterium and can be isolated from freshwater sediments (Allen et al., 2006). Rhodobacter and Pseudomonas, widely distributed in seawater and fresh water, were also found in the intestinal microbiota of Potamopyrgus antipodarum, a freshwater snail from New Zealand (Vesbach et al., 2016). Vesbach et al. (2016) determined that Rhodobacter colonized the intestinal tract of Potamopyrgus antipodarum and formed symbionts with the host. Symbiosis of Rhodobacter with host was also observed in sponges (Halichondria Panicea) and Daphnia (Althoff et al., 1998; Qi et al., 2009).

There were 26 genera significantly increased in HV snails compared to NHV snails, which were potentially plant-related intestinal microbiota, such as Cloacibacteria, Pseudomonas, Flavobacterium, Mycobacterium, and Rhodobacter. Pseudomonas has been identified as a cellulolytic species in invertebrates (Huang, Sheng & Zhang, 2012). In fact, most plant-related bacteria varied with the snail species. Cardoso et al. (2012) reported that the bacterial taxa closely related to herbivores are Pseudomonas, Clostridiaceae, Lactococcus, Bacteroides, Flavobacteriaceae, Mucilaginibacter, Citrobacter, Klebsiella, Aeromonas, Acinetobacter, and Comamonas. Among these bacteria, Pseudomonas, Flavobacteriaceae and Aeromonas were found in our study. Microbacterium, Cellulosimicrobium, Nocardiopsis, Aeromonas, Flavobacterium, Klebsiella were isolated from the intestines of Achatina fulica with cellulose degradation activity (Pinheiro et al., 2015), and in which only Aeromonas and Flavobacterium were found in our study.

In this study, the intestinal microbiota of freshwater snails was modified by diet, which was also reported in other animals, such as mammals (Muegge et al., 2011) and fish (Kokou et al., 2019). A similar study reported that the sugarcane-based diet altered the gut microbiota of snail Achatina fulica (Cardoso et al., 2012). Although Aeromonas was abundant (exhibiting an abundance of 85.40%) in NHV and was scarce in HV (1.78%), this result does not mean that Aeromonas was incapable of digesting plant fiber. Pinheiro et al. (2015) found that the 6 Aeromonas isolated from snail Achatina fulica all had the ability to hydrolyze lignocellulose. Similar conditions were found in the genus Comamonas, which also exhibited significantly higher abundance in NHV than in HV snails, but relevant studies also indicate that it may be related to herbivorous species (Cardoso et al., 2012).

However, more species in NHV snails (154 genera) disappeared or decreased their abundance significantly compared to HV snails, indicating that these species were better adapted to the intestinal environment of herbivorous diet than the intestinal environment of non-herbivorous diet. Little is known about the role of this microbiota in, for instance, assisting the host with food digestion or immunity. The variation of these gut bacteria will provide better insights into the interaction of snails and their gut microbiota.

PICRUSt analysis showed that different feed treatments had a significant effect on the function of snail intestinal microbiota. Most COG functions had significant differences between the HV and the NHV snails. Further KO analysis showed a significant difference in the gene abundance of lignocellulose-active enzymes between the two groups. The abundance of these lignocellulose-related genes was mostly higher in the HV group than in the NHV group, indicating that the intestinal microbiota in herbivorous snails were more effective in at lignocellulosic metabolism than those in non-herbivorous snails. A large number of genes related to protein degradation were higher in HV than in NHV snails, indicating a more efficient protein utilization in herbivorous feeding snails, and this will be helpful to develop microbial proteases from animal intestine.

Conclusions

Our study provides the first characterization of gut community diversity of Planorbella trivolvis, providing insight into gut community structure within freshwater snails. Compared with herbivorous feeding (HV), although non-herbivorous feeding (NHV) promoted the growth of snails, it caused an accumulation of lipid droplets in intestinal mucosal cells. A set of 22 core bacteria was determined to be consistently associated in relatively high abundance (>1%) with the diversity of Planorbella trivolvis. Compared with herbivorous feeding, non-herbivorous feeding reduced the alpha-diversity of the intestinal microbiota of snails and changed the composition of intestinal bacterial communities. For example, the abundance of many bacteria decreased or disappeared. Functional prediction showed that the abundance of cellulose degradation genes in the intestinal microbiota of HV snails was considerably higher than that of NHV snails. In summary, our findings showed novel snail-microbe associations and, furthermore, suggest that the variety of bacteria within the gut might promote a better digestion ability of the host to different diets.

Supplemental Information

Supplemental Information 1 LEfSe analysis of intestinal microbial flora composition and abundance with LDA threshold >4.0.

The HV group is represented by red bars (H) and the NHV group by blue bars (N).

Click here for additional data file.

Supplemental Information 2 The species significant difference between groups distinguished by LEfSe analysis at genus level (LDA threshold >2.0).

Click here for additional data file.

Supplemental Information 3 Snail growth performance.

Click here for additional data file.

Supplemental Information 4 Diets’ nutritional ingredients.

Click here for additional data file.

Supplemental Information 5 R scripts used for the statistical analysis with the vegan package.

Click here for additional data file.

Additional Information and Declarations

Competing Interests

Author Contributions

Data Availability

The authors declare that they have no competing interests.

Zongfu Hu conceived and designed the experiments, performed the experiments, authored or reviewed drafts of the paper, and approved the final draft.

Qing Tong performed the experiments, analyzed the data, prepared figures and/or tables, and approved the final draft.

Jie Chang performed the experiments, analyzed the data, authored or reviewed drafts of the paper, and approved the final draft.

Jianhua Yu analyzed the data, prepared figures and/or tables, and approved the final draft.

Shuguo Li analyzed the data, prepared figures and/or tables, and approved the final draft.

Huaxin Niu conceived and designed the experiments, prepared figures and/or tables, authored or reviewed drafts of the paper, and approved the final draft.

Deying Ma conceived and designed the experiments, authored or reviewed drafts of the paper, and approved the final draft.

The following information was supplied regarding data availability:

The raw reads are available at the NCBI Sequence Read Archive (SRA): PRJNA640745. Raw data for chemical characteristics of alfalfa & growth performance of snails are available in the Supplemental Files.

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
