# Peer review of "Gut bacterial communities in the freshwater snail Planorbella trivolvis and their modification by a non-herbivorous diet"

_PeerJ, doi:10.7717/peerj.10716_

## Round 0.1 · original submission · Major Revisions

Dear Dr. Hu and colleagues:

Thanks for submitting your manuscript to PeerJ. I have now received three independent reviews of your work, and as you will see, the reviewers raised some concerns about the research. Despite this, these reviewers are optimistic about your work and the potential impact it will lend to research studying the microbiome dynamics of Planorbella trivolvis. Thus, I encourage you to revise your manuscript, accordingly, taking into account all of the concerns raised by the reviewers.

Importantly, please ensure that an English expert has edited your revised manuscript for content and clarity. Please also ensure that your figures and tables contain all of the information that is necessary to support your findings and observations. The Methods should be clear, concise and repeatable. Please ensure this. Please address all of the typos.

Please justify your rationale for specific methodologies implemented in your study. Major concerns were raised over this, and your rebuttal should address these issues.

Overall, the quality of the writing is poor and your manuscript should be improved to present the study’s question, framework and outcomes in a more clear and cohesive manner.

There are many minor problems pointed out by the reviewers, and you will need to address all of these and expect a thorough review of your revised manuscript by these same reviewers.

I agree with the concerns of the reviewers, and thus feel that their suggestions should be adequately addressed before moving forward. Therefore, I am recommending that you revise your manuscript, accordingly, taking into account all of the issues raised by the reviewers.

I look forward to seeing your revision, and thanks again for submitting your work to PeerJ.

Good luck with your revision,

-joe

Reviewer 1 ·

Basic reporting

The manuscript suffers from a lack of clarity due to poor use of English.

Experimental design

I outline the issues that I detected here in the general comments below.

Validity of the findings

No comment.

Additional comments

The manuscript by Hu et al. describes the intestinal microbiota of the freshwater snail Planorbella trivolvis and its modification by an artificial diet. There is merit in the data as there is still scarce information about snail microbiota. Snail intestinal microbiota is of interest not only for basic scientific knowledge but also because it harbors bacteria that can process plant biomass and digest cellulose and lignocellulose, and the enzymes of these bacteria could have biotechnological potential in a variety of industrial processes. The authors used Illumina sequencing of the V3-V4 region of the 16S rRNA gene to study the taxonomic composition of the intestinal microbiota and analyzed it with established methodologies. I think that the work could be published as long as the authors address several issues that I will outline below:

General comments:
The manuscript suffers greatly from a lack of clarity due to many mistakes in the use of the English language. The authors should send their work to an expert editing service to substantially improve readability and clarity.

Specific comments:

Introduction:

1- Please do not refer to mollusks in a scientific paper as shellfish that is a generic term used for a variety of different animals. Call snails, snails, and slugs, slugs, for example.
2- On line 94, the authors mention recent studies that investigated the intestinal bacterial communities of freshwater snails but do not provide a reference.
3-The authors did not give a specific reason for analyzing the effects of a modified diet on the intestinal microbiota. Is there a specific reason for this or a proposed hypothesis?

Materials and methods:

1-Where were the snails obtained? Were they collected in natural environments? Were they part of an established colony?
2-What do you mean by three repetitions? (line 106)
3-Market purchased feed is not clear (line 108). What is the brand? What kind of feed is it? Is it for fish feeding?
4-I understand the need for pooling the intestines but why did the authors only used 5 pooled samples (that would account for 25 snail intestines in total, 5 intestines per analyzed sample). The authors should have a really good reason for not analyzing more samples. At least 10 samples for each experimental condition would be the minimum number for a proper analysis. That would only require 50 snails in total for each condition. Illumina sequencing would allow the simultaneous analysis of dozens of samples so there were no limitations regarding the sequencing run.
5-Please describe the primers used for the V3-V4 region and give a suitable reference or describe how they were designed if this was done by the authors themselves. Please indicate the size of the amplified product.
6-How many PCR reactions were performed for each sample?
7-I do not understand the need for a library prep kit (line 142) as the 16S rRNA amplicons are usually directly sequenced. Please explain.
8- Please supply the R scripts used for the statistical analysis with the vegan package as supplementary material.

Results:

1-OTUs were consistently misspelled as OUT.
2-On line 256, the authors state that phylogenetic analysis showed that genera that were present in a higher proportion in the HV group were closely related but they did not provide the data or figure.
3-On line 269, the authors describe LEfSe analysis but there is no description on how it was done in the materials and methods section. Please provide this information.
4-From line 309 to the end of the paragraph the text is of poor clarity and it needs to be improved to increase readability.
5-As it was done for enzymes related to plant degradation it would be of interest to perform this analysis for enzymes related to protein degradation such as proteases and peptidases.

Reviewer 2 ·

Basic reporting

The MS by Hu and co-workers describes the differences in the microbiota of one species of freshwater snail (Planorbella trivolvis) in relation to diet (plant-based vs. commercial pellet). In this context, authors carried out a targeted amplicon-sequencing of 16S rRNA bacterial genes to delineate differences in composition, richness and diversity of bacterial communities in the gut of the two groups of snails and also studied the predicted metagenome by using PICRUST. Authors found some differences in the studied variables but results are confusedly presented and discussed. In my opinion the MS is not suitable for publication in its current form. The introduction and discussion sections are week and contains many flaws and confusing terms.

The use of English language is limited and careless. In my opinion, the MS needs a thoughtful language edition by someone with a full proficiency in English. The MS contains many colloquial expressions, confuse terms and speculative statements. Overall, the MS needs a thoughtful revision to improve in clarity and readability.

Experimental design

Despite the research is within the scope of the journal, the research question is not well defined and authors need to highlight why it is important the study of the microbiota of this snail species.

The experimental design has flaws and many details are missing in the Material and Methods section (too numerous to cite here but see some in the general comments section). Some sections of the M&M are especially troublesome and confuse.

Validity of the findings

Authors found some differences in the studied variables and also in the predicted abundance of some genes encoding cellulolytic enzymes. Despite these findings, results are confusedly presented and discussed. Since many methodological details are missing the statistical significance of the findings is difficult to assess.

Additional comments

The term “intestinal (or gut) flora” is outdated and should be replaced by “intestinal (or gut) microbiota” throughout the MS.

Some specific comments are highlighted below:

L26: High compared to what? To NHV snails? How high is high?
L28: intestinal flora is an outdated term, I suggest using “intestinal micorbiota” throughout the MS
L28: herbivorous gut of what type of animals? mammals? other?
L31: These are phylums… this sentence provides no information… of course there will be differences in members of these large phyla. Be more precise. By the way, Bacteroidetes is misspelled.
L37-39: Well, ok but why it is important to study the gut microbiota of these snails? Authors should sell better their story to highlight the importance of their findings.
L41: Keywords is misspelled
L412: the last two keywords are not keywords. Please, delete
L46: I guess that citing the genus here is dispensable.
L47: Species names should be written in italics and in small caps for the species designation (Planorbella trivolvis)
L50: Ribeiroia ondatrae should be written in italics
L57: “a gut microbiome specialized in the…”
L58: Maybe a reference is needed here to support this argument
L59: I suggest rewritting this final sentence here. Write more concisely.
L45: “Planorbella trivolvis belongs to…. I suggest writing more concisely.
L51: “huge ecosystem of bacteria”? Do you mean a huge bacterial diversity in the intestinal tract of animals…?
L65: Again, I suggest getting rid of this outdated term throughout the MS (flora)
L67: “…to efficiiently digest…”
L71: “no dominant phenomenon of certain strains”? What does this means? Please, be precise when writing
L72: replace “flora” by “microbiota”
L79: “Are relatively more” what? Please, be precise when writing.
L83: “functional intestinal microorganisms”? What is that? Again, be precise when writing
L84: Excavation? I’m lost...
L85: delete “snail” as adjective for “metagenomic study”
L87: GH? Please, define
L89: As simply as that? Again, be precise when writing.
L90: Replace “flora” by “microbiota”
L91: A single gene or various? Please, rewrite.
L92: See above comments on the use of “flora”
L93: developing enzymes??
L94: If there are recent studies, please cite them
L99: under conditions of oblivion? Again, I’m lost.
L100: to represent?
L105: many key details are missing: How many animals in total? How many by group?
L112: This part is specially confusing. Also, it is written in present tense instead of past tense.
L122. This section needs explanation.
L129: All together? By individual?
L130: Authors extracted all DNA not only bacterial genomes.
L131: There are many DNA extraction kits by MP-Bio, please, be more precise.
L132: OD? Plese, define
L133: Add reference to the Company.
L134: which primers?
L149: Reference to this software?
L152: Please, cite references by Edgar as author of UCHIME and Uparse, not the web address.
L153-154: Cte QIIME authors.
L163: performed is misspelled.
L165: Analysis of alfa diversity indicators is not statistics but analysis of sequence datasets. Also, please, cite Schloss and co-workers when citing Mothur.
L204-205: 307+90 = 397… but authors stated that they delineated 525 OTUs, so, where are the other 228 OTUs?
...

Reviewer 3 ·

Basic reporting

Clarity and quality of the writing needs to be improved. Data has been shared with NCBI. See general comments.

Experimental design

Original research. The knowledge gap could be more clearly defined. Overall, seems to be of high quality, but more details on methods needed. See general comments.

Validity of the findings

Findings seem to be generally valid. Conclusions could be more effectively summarized. More detail on data collection is needed.

Additional comments

Summary – Hu et al. describe an experiment where freshwater snails (Planorbella trivolvis) were fed either herbivorous (HV) or non-herbivorous (NHV) feed. Their main focus was on comparing the microbial community of snails fed the different diets. They also examined differences in snail growth and intestinal morphology, as well as potential differences in metabolic functioning; they found that diet had clear effects on bacterial communities and snail traits.

General comments
The study provides new insights into how diet affects the gut microbiome of these snails and other effects of diet. However, the quality and clarity of the writing needs to be improved. There are many typos and grammatical errors, particularly in the Introduction and Discussion, which in some places make it difficult to understand what the authors are trying to communicate. I point out a few examples below, but there are many more. Careful copyediting is thus needed. The introduction should also be revised to more clearly set up the objectives of the paper, and the discussion could be more focused on discussing rather than rehashing the results. I also think some important methodological details are currently lacking from the Methods (see specific comments below).

Specific comments

Abstract
Line 12 – “invasive” depends on the context? In China they are invasive? They are native many other places.
Line 13 – “in aquarium of ornamental fish market” – this phrasing doesn’t make sense. This is one of a number of sentences that need to be rewritten. I point this out as an example, but there are many others with incorrect or confusing phrasing.
Line 13-14– Unclear what “strong adaptability to food” means. Also, the subject “Planorbella trivolvis is singular, but the verb “feed” is plural. Again, this is one example of an error that occurs repeatedly – subject-verb agreement needs to be checked throughout the paper.
Line 15 – Another example - “which allows them to cross the oversea to China by Ornamental fish trade” – not grammatically correct and confusingly written. Also, “ornamental” does not need to be capitalized (see, e.g., lines 298-306 for other words that are capitalized but don’t need to be).
Line 19 – “The variation of intestinal flora from herbivorous feed (HV)” – again, confusing phrasing. As phrased, the text indicates the focus was on the flora in the feed. It needs to specify that in fact the authors examined the flora in the snails fed different diets.
Line 21-22 – I’m not sure this level of detail is needed in the abstract – the previous sentence already indicated that the main treatment was that snails were fed different diets.
Line 34 – “shifted to” is not really correct. Perhaps, “contrasted with”?
Line 35 – “Functional prediction” – unclear what this means. Also, it would help to explain what COG functions refers to.

Intro
Line 45 – “Planorbella trivolvis is taxonomic belonging to the family Plamorbidae” – unclear
Line 47, 50 (and elsewhere) – Species and genus not italicized
Line 49 – “Trematode” misspelled. This is one of several examples (e.g., perofrmed in line 163, sanils in line 203, miacrobial in line 327) that would be caught by a spell-check.
Lines 61-93– These paragraph could be more focused to outline specifically which gaps occur in knowledge of animal gut microbiomes, which would motivate the work done for this study.
Line 66 – Seems to be redundant with line 58.
Line 94-96– Unclear what the point of this paragraph is.
Line 98-99– “which has shown remarkable food adaptation and survival under conditions of oblivion” – unclear
Line 100 – “represent” is wrong word here
Line 97-102 – This paragraph seems to be explaining the main objectives of the study. Would an additional objective be examining effects of diet on snail growth and intestinal morphology? Also to examine functional differences in the microbiome of snails fed different diets? I think it needs to be clear later on in the paper that those were also objectives, or else it is unclear why those analyses were done.

Materials and Methods
Lines 104-111– Details are needed. What was the origin of the snails used? Were they reared from eggs? If not, what were they fed prior to the study? Were snails that were ultimately used for different samples housed separately? If not, then there may be an issue of pseudoreplication (e.g., if all the NHV group were housed in the same aquarium, then different snails would not be truly independent from each other). What is pellet feed composed of?
Lines 114-117– Need to be in past tense. Phrased as instructions, rather than describing what was done.
Line 122-127– It’s odd to just list these formulas without context. A sentence or two is needed introducing why these formula are being used and what they are.
Line 129 – It’s unclear what the mucosal samples are and why the intestinal contents were mixed with them.
Line 130 – Rephrase to clarify the MP BIO kit was used to do the extraction according to the kit instructions. As written, it doesn’t actually say that the kit was used, just that the instructions were followed.
Line 134 – What specifically were the primers? Is there a citation? Also, the tense here is wrong.
Line 149 – Is a reference needed for Trimmomatic?
Line 158 – Unclear what PICRUSt is – a brief explanation may help.
Line 170-177 – It seems like this belongs earlier, maybe before explaining the molecular work. As I note above, it would be good to explain why intestinal morphology was examined in the Introduction – how does it fit into the overall objectives of the paper?

Results
Lines 180-191- Did the authors test if the initial size/mass of the snails in each treatment group differed at the beginning of experiment? It would be good to know that initial size did not differ.
Lines 192-197– The authors haven’t given much context for these results. It should be clear why the histology was done, and what the differences between treatment groups were, and what those differences mean.
Line 206-207– Rather than “belong to”, I would say the OTUS “include members of” the different taxonomic groups listed. “All of the alpha-diversity indices” – would be good to list them here.
Lines 210-213 - “the rarefaction curves showed clear asymptotes . . . further sequencing would be unlikely to significantly increase the observed microbial diversity detected” – I disagree – they clearly don’t actually fully level off, and the number of OTUS would continue to increase with more samples. But, I don’t think it really matters. What matters is that, for a given level of sampling effort, there are clear differences between the treatment groups. But I don’t think this sentence is accurate as written.
Lines 226-251: It seems like this could be more succinct and summarize the key patterns that are shown in Fig. 3. Overall, I think there are several places in the Results that just seem to be listing taxa (e.g., line 287-292) or other information (e.g., functions in lines 298-303) that is already effectively summarized in a figure.

Discussion
Lines 340-342: “it was found in experimental procedures that the intestines of NHV snails became easily broken” – where is this reported in the Results?
Lines 342-344: “Second, HE staining of intestinal sections of the NHV group showed that
the fat droplets in the epithelial columnar cells had increased, and striated border
detached.” – which is a measure of intestinal health? and is different that the HV snails?
Line 345-353 – seems to be redundant with the Results. x
Line 361 – “the regulaton of feed” – unclear what this means
Line 362 – “through the feed treatment experiment” – this clause isn’t needed
Line 367-368 – “A large number of Proteobacteria, Bacteroidetes and Acidobacteria” – a large number of bacteria? Or is this referring to high relative abundance? The phrasing is unclear.
Lines 373-379 – Again, seems to be just restating rather than discussing the Results. Also lines 404-408
Line 379-380 – “Due to the limitation of determination methods, many previous studies have reported less at the genus level.” – unclear
Line 384-387 – “Many of the genera in our study were not reported in previous studies, indicating that the intestinal flora composition of Planorbella trivolvis differs from that of other terrestrial or aquatic gastropods.” – this is an overstatement. Since different methods were used, it is important to be cautious in comparing across studies.
Lines 388-403 – The flow of this paragraph, and the links between the different pieces of information, need to be improved.
Lines 397-399 – unclear what this is saying.
Lines 410-419 – Here and elsewhere, reports of results of other studies are described, but it is not clear how findings from these other studies relate to the results of this study.
Line 421 – the reference to “Termitomyces association” is confusing – what is this referring to?
Lines 421-451 – This paragraph could be more concise
Line 466 – “across the phylogenetic diversity of the Planorbella trivolvis” – unclear phrasing
Table 1 – The title of the table only says alfalfa, but both food items are shown. Where did the data on the composition of the feed come from? From the manufacturer?

Table 2 – I suggest putting the letters (a or b) in a different column, or just note that the treatment groups differed for all metrics.

Table 3 – “Indices” rather than “Indexes”. Are p-values from a t-test?

---

## Round 0.2 · Minor Revisions

Dear Dr. Hu and colleagues:

Thanks for revising your manuscript. The reviewers are mostly satisfied with your revision (as am I). Great! However, there are a few issues still to entertain. Please address these ASAP so we may move towards acceptance of your work.

Best,

-joe

Reviewer 1 ·

Basic reporting

no comments

Experimental design

no comments

Validity of the findings

no comments

Additional comments

no comments

Reviewer 3 ·

Basic reporting

See general comments. Some of the writing could be improved for clarity and to better explain how this study fits into the literature.

Experimental design

No comment.

Validity of the findings

No comment.

Additional comments

General Comments
Overall, most of my previous concerns have been addressed, but the writing could still be improved. Results could be more contextualized in existing literature to spell out how this study fits in and fills an existing knowledge gap. Information from other studies in the Intro and Discussion is provided without context; the outside literature should be more effectively integrated into the paper. Generally, the flow and clarity of the writing could still be improved as well.

Specific comments
Title - I suggest removing “and their functional prediction” from the title. It’s not clear what “functional prediction” refers to.

Line 15-16: It’s unclear what “without specific feeding” means. I suggest removing or revising.

Lines 21 – I would revise this sentence to move or remove “After 60 days of rearing” – that clause does not modify “the results”, so should go elsewhere in the sentence.

Line 35 – “different” should be “difference”

Line 48 – “Trivolvis” should not be capitalized.

Line 51 – “Echinostome” should be “echinostomes”

Lines 46-57 – The writing here and elsewhere in the paper (particularly portions of the Intro and Discussion) is choppy. For example, “Snails are common in ponds, lakes and marshes. Snails feed on algae, aquatic plants, the fallen leaves of terrestrial plants and various types of detritus.” Writing like that is listing information without connecting the ideas. Use of transition words or other revisions could help improve the flow between those two (and other) sentences. I suggest revising the text throughout the manuscript to improve the flow of the writing.

Line 74 – “Two articles examined the gut microbiota of freshwater snails.” – Why did the authors choose to focus on these two articles? Are they the only two prior studies on this topic? I suggest revising to clarify.

Line 74-100 – I’m not sure the details from these other studies are necessary to include or useful to the reader. I think what is important is to spell out specifically what knowledge is missing in the literature, and the gaps that the current study sought to address.

Line 101 – I would either revise or add a topic sentence that introduces the focus of this paragraph. I think the specific goals of this study merits its own paragraph.

Line 117 – “When sufficient number” should be “When a sufficient number”

Line 118 – “newborn snails were prepared” - confusing phrasing. Prepared for what?

Line 117-120 – I would revise this sentence – it is awkwardly phrased.

Line 127 – Awkward sentence.

Line 132-33 – “The shells were carefully destroyed and disassembled from the snail body.” – Destroyed? Is that the right word. It’s unclear what that means.

Line 213 – I suggest changing “Growth of snail” to “Snail growth”
Line 215-217: I think the numbers in the text aren’t necessary, since they are provided in the table.

Line 233 – Missing a period somewhere.

Line 234 – “in this image” – in what image? Should a figure be indicated?

Line 237 – “These differences in the results” – Which differences specifically? Phrasing is vague.

Line 253 – “good’s” should be capitalized

Line 281 – missing a space “groups.Others”. Noticed at least a couple other times in the manuscript (e.g., line 288, 384), so would check throughout.

Lines 292 – It’s unclear what “Closer analyses” means. I would revise to clarify.

Line 350 – “This may be affected” Awkward phrasing.

Line 366-69 – It’s unclear why the authors decided to compare snails with bison. Is there a reason to compare these two taxa? If so, the reasoning should be clarified.

Line 370-71 - “Core microbes were considered to be diet generalists under the condition of high variation of diets.” It’s unclear what this sentence is saying. I suggest revising.

Line 372-77 – It’s very unclear what the point of the comparison with these other studies is or what the author’s point here is.

Line 378-79 – Unclear what “in the water environment” means? Free-living in the pond water itself? Or in aquarium water?

Line 401-403 “In this study, the intestinal microbiota of freshwater snails was modified by diet, as well as other animals, such as mammals (Muegge et al., 2011) and fish
403 (Kokou et al., 2019).” – Confusingly written. As written, suggests other animals were included in this study. I know that’s not what the authors actually mean, but the text needs to be revised to actually reflect what the authors are trying to say.


“A similar result reported” – A result reported? I think there needs to be another subject to this sentence.

Line 418 – “snail” should be “snails”

Line 429 – “this will be help” – needs to be revised for grammar

Line 446 – “better adaptation” – adaptation is the wrong word in this context

---

## Round 0.3 · accepted · Accept

Dear Dr. Hu and colleagues:

Thanks for again revising your manuscript. I now believe that your manuscript is suitable for publication. Congratulations! I look forward to seeing this work in print, and I anticipate it being an important resource for research studying the microbiome dynamics of Planorbella trivolvis. Thanks again for choosing PeerJ to publish such important work.

Best,

-joe